# The Known Unknowns: An Enigmatic Pathway of C_17_-Polyacetylenic Oxylipins in Carrot (*Daucus carota* L.)

**DOI:** 10.3390/cimb47060471

**Published:** 2025-06-19

**Authors:** Abdul Wakeel Umar, Hamad Hussain, Naveed Ahmad

**Affiliations:** 1BNU-HKUST Laboratory of Green Innovation, Advanced Institute of Natural Sciences, Beijing Normal University at Zhuhai (BNUZ), Zhuhai 519087, China; awzju@yahoo.com; 2Department of Agriculture, Faculty of Chemical and Life Sciences, Abdul Wali Khan University Mardan, Mardan 23200, Pakistan; hammadagarian629@gmail.com; 3Guangdong-Hong Kong Joint Laboratory for Carbon Neutrality, Jiangmen Laboratory of Carbon Science and Technology, Jiangmen 529199, China

**Keywords:** C_17_-polyacetylenic lipids, falcarindiol, sensory properties, biosynthesis, *Daucus carota* L.

## Abstract

C_17_-polyacetylenic (PA) oxylipins are bioactive compounds in carrots (*Daucus carota* L.) with structurally unique features and diverse biological roles. These PA-derived compounds have garnered attention for their potential contributions to human health, particularly in cancer prevention and anti-inflammatory applications. This trade-off between health benefits and sensory quality underscores the importance of understanding the genetic and biochemical basis of PA biosynthesis, as it may allow for the development of carrots with optimized levels of these compounds that balance both nutritional and sensory qualities. In this review, we seek biochemically inspired strategies to elucidate the complexities of PA-derived oxylipins biosynthesis in carrots, a topic that remains largely unexplored. By integrating current knowledge on polyacetylene biology, biosynthesis, genetic and enzymatic factors involved in their production and the implications for enhancing the medicinal value of carrots we aim to provide a foundation for future research that could unlock the full potential of carrots as a source of health-promoting bioactive compounds.

## 1. Introduction

Polyacetylenes (PAs) are a class of chemical compounds characterized by the presence of multiple carbon–carbon triple bonds [1]. These PA-derived oxylipins are notable for their diverse biological activities, including anti-inflammatory, antimicrobial, and anticancer properties, making them significant for both health and commercial applications. Specifically, falcarinol has emerged as a key compound in cancer research, where its ability to hinder the proliferation of various cancer cell types—including those from breast, colon and leukemia origins—has been documented [2,3,4]. Studies indicate that falcarinol primarily induces apoptosis and disrupts the cell cycle, thereby constraining tumor cell growth at multiple stages in the malignant process [1,5]. Falcarindiol, on the other hand, has garnered interest for its pronounced anti-inflammatory capabilities. Research shows that falcarindiol modulates the inflammatory response by inhibiting key pro-inflammatory cytokines and enzymes, most notably cyclooxygenase-2 (COX-2) [5,6]. This modulation is vital, as COX-2 is a known contributor to inflammatory diseases and cancer progression [7]. Furthermore, falcarinol and falcarindiol compounds have been shown to enhance glucose uptake in muscle tissues, suggesting their utility in metabolic health, thereby linking the consumption of carrots and similar vegetables directly to health benefits [8]. The relevance of such PAs extends beyond the laboratory setting into practical applications, including their integration into nutraceuticals, functional foods, and possible therapeutic agents against chronic diseases. Hence, understanding the stereochemistry and pharmacodynamics of these compounds is paramount, as various configurations can yield differing bioactivities and efficacy [9,10].

Falcarin-type PAs are predominantly found in members of the Apiaceae family, which includes economically important plants such as carrots (*Daucus carota* L.), celery (*Apium graveolens* L.) and parsley (*Petroselinum crispum* L.) [11]. In particular, falcarinol and falcarindiol have been identified as the most abundant classes of PAs in the carrot species [1]. However, the production of these bioactive compounds can vary due to factors such as cultivation method, genetic factors and environmental conditions, highlighting the need for optimized agricultural practices to maximize their concentrations in food sources [12,13]. For instance, wild carrot varieties tend to have higher levels of PAs compared to domesticated cultivars, which are often bred for traits such as sweetness and reduced bitterness rather than for bioactive compound content [14]. Additionally, environmental factors such as soil type, temperature and light exposure can influence the expression of genes involved in PAs biosynthesis, thereby affecting the concentration of these compounds in the edible parts of the plant [15]. The variability in PA content among different carrot genotypes and growing conditions poses both opportunities and challenges. On one hand, it offers the potential for breeding or genetically engineering carrot varieties with enhanced levels of PAs, which could be marketed as functional foods with added health benefits. On the other hand, this variability complicates efforts to standardize carrot-derived products, such as juices, extracts and supplements, for consistent therapeutic use. The lack of a clear understanding of the biosynthetic pathway also limits the ability to manipulate PA levels through targeted breeding or genetic modification. Despite their promising biological activities, the pathways leading to the biosynthesis of PAs in plants, particularly in carrots, remain enigmatic.

Current knowledge indicates that PAs are synthesized from fatty acid precursors, specifically oleic and linoleic acids, through a series of desaturation and acetylenation reactions [15,16,17] (Figure 1). These reactions are catalyzed by enzymes such as fatty acid desaturases (FADs) and acetylenases, which introduce triple bonds into the carbon chain, converting fatty acids into PAs compounds [18,19,20]. However, the specific enzymes involved in the desaturation and acetylenation steps leading to falcarinol and falcarindiol production in carrots have not been fully identified. Moreover, the regulation of these biosynthetic pathways at the genetic and epigenetic levels is poorly understood, posing a significant challenge to the development of strategies aimed at enhancing PA content in crops. In this review, we aimed to present a comprehensive overview of the current knowledge on the biosynthesis of C_17_-PAs in carrots, while also identifying the key gaps in our understanding that warrant further investigation. First, this review will explore the biological significance of common PAs such as falcarinol and falcarindiol, providing further insights into their known mechanisms of action and potential health benefits. Second, the review will address the biosynthetic pathway of PAs in carrots, with a particular focus on the enzymatic and genetic factors involved in the formation and molecular regulation of falcarinol and falcarindiol. It will also examine the influence of environmental factors on PA biosynthesis, emphasizing the need for more research into the interaction between genotype and environment in determining PA content. Third, the review will conclude by outlining the key research questions that remain unanswered in the study of C_17_-PA derived oxylipins in carrots as well as propose future research directions aimed at addressing these gaps, with the goal of advancing our understanding of this enigmatic pathway and its potential applications in agriculture and human health.

## 2. Chemical Nature and Biological Importance of C_17_-Polyacetylenes

C_17_-PA compounds represent a structurally distinct class of oxylipins derived from crepenynic and dehydrocrepenynic acids, which are classified as “unusual” polyunsaturated fatty acids. From the vast array of over 1400 polyacetylenes documented in higher plants [22], a unique group of 12 structurally interconnected PAs was extracted from carrots, purified and their molecular architectures elucidated (Table 1). The aliphatic PAs such as falcarinol, falcarindiol and falcarindiol-3-acetate were isolated from carrot, each characterized by dual double bonds at C1/2 and C9/10, paired with two triple carbon-carbon bonds at C4/5 and C6/7, alongside a seven-carbon aliphatic tail (C11–C17) [23]. Falcarinol possesses a single hydroxyl group at C3, whereas falcarindiol introduces an additional hydroxyl at C8. In contrast, falcarindiol-3-acetate features an acetyl group at C3. Beyond these dominant PAs, nine other bisacetylenes were recently identified in carrots, including (E)-isofalcarinolone, falcarindiol-8-acetate, 1,2-dihydrofalcarindiol-3-acetate, (E)-falcarindiolone-8-acetate, (E)-falcarindiolone-9-acetate, 1,2-dihydrofalcarindiol, (E)-1-methoxy-falcarindiolone-8-acetate,(E)-1-methoxy-falcarindiolone-9-acetate and panaxydiol [2,23].

The levels of PAs in carrots are shaped by a variety of factors, including the carrot variety, plant age, developmental phase, root size, storage duration and the geographical region of cultivation [[2],,[24]]. Additionally, both biotic and abiotic stressors, encountered during field growth or postharvest storage, play a significant role in modulating PA concentrations [25,26]. For instance, falcarindiol content in cultivated orange carrots stands out as the most prevalent PA, with levels ranging from 16 to 84 mg/kg fresh weight (FW), followed by falcarinol (8–27 mg/kg FW) and falcarindiol-3-acetate (8–40 mg/kg FW). The carrot genotype notably affects the quantities of falcarinol and its derivatives; for instance, a study of 27 carrot cultivars grown under identical conditions showed falcarinol concentrations varying from 7.0 to 40.6 mg/kg FW [27]. Beyond genetic factors, PA distribution within carrot plant organs also differs significantly. Previous studies explored the spatial arrangement of falcarinol, falcarindiol and falcarindiol-3-acetate from the root’s top to bottom and from the outer phloem to the inner xylem [25]. The bitter-tasting upper root end and phloem contained higher levels of falcarindiol-3-acetate (33.5 and 32.3 mg/kg FW, respectively), while the less bitter lower end and xylem had much lower concentrations (1.8 and 1.5 mg/kg FW) [28]. Among bisacetylenic oxylipins, falcarinol and falcarindiol exhibited similar concentrations in the phloem and xylem, but falcarindiol-3-acetate levels in the phloem were twice those in the xylem [17]. Similarly, PAs were found to be concentrated in vascular bundles of the young secondary phloem and in pericycle oil channels near the periderm, suggesting these structures facilitate the transport and storage of PAs [29]. cimb-47-00471-t001_Table 1Table 1List of identified C_17_-polyacetylenes in *Daucus carota*, their associated structural features, fatty acid desaturase (FAD2) biosynthetic genes and potential transcriptional regulators.C17-PolyacetyleneStructural FeaturesStructural GenesTranscription FactorsReferencesFalcarinolOne hydroxyl group at C3, two double bonds (C1/2, C9/10), two triple bonds (C4/5, C6/7)DcFAD2-6 (Δ12 acetylenase), DcFAD2-11 (Δ12 desaturase), DcFAD2-13, DcFAD2-14 (bifunctional 12/Δ14 desaturases)None specifically identified; potential MYB/bHLH involvement speculative[2,20,28]FalcarindiolHydroxyl groups at C3 and C8, two double bonds (C1/2, C9/10), two triple bonds (C4/5, C6/7)DcFAD2-6, DcFAD2-11, DcFAD2-13, DcFAD2-14None specifically identified; potential MBW complex (MYB, bHLH, WD40) involvement speculative[2,20,28]Falcarindiol-3-acetateAcetyl group at C3, hydroxyl at C8, two double bonds (C1/2, C9/10), two triple bonds (C4/5, C6/7)DcFAD2-6, DcFAD2-11, DcFAD2-13, DcFAD2-14None specifically identified; potential MBW complex involvement speculative[2,20,28](E)-IsofalcarinoloneKetone group, two double bonds, two triple bondsLikely derived from FAD2-mediated pathways; specific genes not confirmedNone specifically identified[2,20,28]Falcarindiol-8-acetateAcetyl group at C8, hydroxyl at C3, two double bonds, two triple bondsLikely derived from FAD2-mediated pathways; specific genes not confirmedNone specifically identified[2,20,28]1,2-Dihydrofalcarindiol-3-acetateDihydro at C1/2, acetyl group at C3, hydroxyl at C8, one double bond, two triple bondsLikely derived from FAD2-mediated pathways; specific genes not confirmedNone specifically identified[2,20,28](E)-Falcarindiolone-8-acetateKetone group, acetyl at C8, two double bonds, two triple bondsLikely derived from FAD2-mediated pathways; specific genes not confirmedNone specifically identified[2,20,28](E)-Falcarindiolone-9-acetateKetone group, acetyl at C9, two double bonds, two triple bondsLikely derived from FAD2-mediated pathways; specific genes not confirmedNone specifically identified[2,20,28]1,2-DihydrofalcarindiolDihydro at C1/2, hydroxyls at C3 and C8, one double bond, two triple bondsLikely derived from FAD2-mediated pathways; specific genes not confirmedNone specifically identified[2,20,28](E)-1-Methoxy-falcarindiolone-8-acetateMethoxy group at C1, ketone, acetyl at C8, two double bonds, two triple bondsLikely derived from FAD2-mediated pathways; specific genes not confirmedNone specifically identified[2,20,28](E)-1-Methoxy-falcarindiolone-9-acetateMethoxy group at C1, ketone, acetyl at C9, two double bonds, two triple bondsLikely derived from FAD2-mediated pathways; specific genes not confirmedNone specifically identified[2,20,28]PanaxydiolTwo hydroxyl groups, two double bonds, two triple bondsLikely derived from FAD2-mediated pathways; specific genes not confirmedNone specifically identified[2,20,28]Note: The genetic basis for the structural diversity of PAs and their downstream modifications is not fully elucidated. Further functional genomic studies are needed to identify additional genes and confirm transcription factor roles.

The accumulation PAs in the outer tissue layers of carrot roots aligns with their protective role as an antifungal barrier, particularly in young roots. These compounds are vital for plant defense, shielding against phytopathogenic fungi, nematodes and insects [29]. For instance, elevated levels of PAs like falcarinol have been observed in tomato fruits and leaves under attack by a variety of fungal pathogens. In carrots, falcarindiol has demonstrated notable inhibitory effects against the fungal leaf blight pathogen *Alternaria dauci* [30]. In the partially resistant carrot cultivar “Bolero”, falcarindiol concentrations in leaves were deemed sufficient to suppress *A. dauci* development [30]. Consequently, PAs are recognized as phytoalexins—small molecular weight compounds synthesized by plants in response to microbial assaults or abiotic stresses, such as UV exposure or high salinity [17]. Therefore, strategic breeding initiatives are essential to develop high-yielding, late-bolting wild carrot varieties or to engineer cultivated carrots enriched with PAs, enabling cost-effective, large-scale production of these compounds for pharmaceutical purposes.

## 3. Bioactive Potential of C17-Polyacetylenes

The repertoire of biological and pharmacological properties linked to PAs is expanding, with these diacetylenes recognized as key contributors to the health benefits derived from consuming fruits and vegetables [31]. Notably, aliphatic C_17_-PAs of the falcarinol type demonstrate significant antimicrobial, anti-inflammatory and anticancer effects (Figure 2) [32]. The subsequent section explores the specifics of these pharmacological activities in detail.

### 3.1. Anticancer Activities of C_17_-Polyacetylenes

C_17_-PAs have drawn significant attention due to their potent anticancer properties, with multiple studies highlighting their ability to inhibit the proliferation of cancer cells and induce apoptosis [3,33]. Their anticancer activity is partly attributed to their pro-apoptotic effects, which involve the activation of caspase-dependent pathways [34]. In vitro studies have demonstrated that falcarinol and falcarindiol inhibit the growth of human leukemia, colon and breast cancer cell lines by promoting cell cycle arrest in the G0/G1 phase and triggering programmed cell death [35,36]. The mechanisms behind this include the upregulation of pro-apoptotic proteins such as Bax and the downregulation of anti-apoptotic factors like Bcl-2 [37,38]. This apoptosis-inducing capability makes C_17_-PAs promising candidates for cancer therapy. In addition to apoptosis induction, C_17_-PAs exert anticancer effects by modulating key signaling pathways associated with cancer progression. One of the primary targets is the nuclear factor erythroid 2-related factor 2 (Nrf2) pathway, which regulates the expression of detoxification and antioxidant enzymes [39,40,41]. Falcarindiol, in particular, has been shown to activate the Keap1-Nrf2 pathway, enhancing the production of phase II detoxifying enzymes, such as glutathione S-transferases (GSTs) and NAD(P)H quinone oxidoreductase 1 (NQO1) [42]. This mechanism reduces oxidative stress and counteracts the formation of reactive oxygen species (ROS), which are implicated in carcinogenesis. Moreover, C_17_-PAs can suppress cancer cell migration and invasion by downregulating matrix metalloproteinases (MMPs) [43,44], thereby reducing metastatic potential.

The anticancer effects of C_17_-PAs also involve modulation of nuclear receptors, such as peroxisome proliferator-activated receptor gamma (PPARγ) [45]. By acting as partial agonists of PPARγ, falcarinol and its derivatives inhibit cancer cell proliferation and promote differentiation, suggesting that PPARγ activation is associated with reduced tumor growth and enhanced sensitivity to chemotherapy [12]. These crucial insights indicated that C_17_-PAs could serve as adjuvants to conventional anticancer drugs, improving their therapeutic efficacy. Furthermore, in vivo studies have demonstrated that regular consumption of PA-rich carrots reduces the incidence of colorectal cancer in animal models, underscoring their dietary chemopreventive potential [46,47].

### 3.2. Anti-Inflammatory Properties of C_17_-Polyacetylenes

The anti-inflammatory activities of C_17_-PAs are well-documented, with their ability to suppress pro-inflammatory mediators and modulate immune responses [1,7,48]. Falcarinol and falcarindiol inhibit the production of key inflammatory cytokines, such as interleukin-6 (IL-6), tumor necrosis factor-alpha (TNF-α) and interleukin-1 beta (IL-1β), in activated macrophages [6,49]. These cytokines are major drivers of chronic inflammation and are implicated in the pathogenesis of diseases such as arthritis, atherosclerosis and inflammatory bowel disease (IBD). By attenuating the release of these mediators, C_17_-PAs demonstrate potential as natural anti-inflammatory agents [7,50]. At the molecular level, the anti-inflammatory effects of C_17_-PAs are partly mediated through the inhibition of cyclooxygenase (COX) and lipoxygenase (LOX) enzymes [1]. Prior studies have shown that falcarinol and falcarindiol inhibit COX-2, the enzyme responsible for the synthesis of pro-inflammatory prostaglandins, thereby reducing inflammation and pain [1,33].

Additionally, they suppress the activity of 5-, 12- and 15-LOX, preventing the formation of leukotrienes, which are potent inflammatory mediators. This dual inhibition of both COX and LOX pathways makes C_17_-PAs effective in reducing inflammation through multiple mechanisms, offering advantages over synthetic nonsteroidal anti-inflammatory drugs (NSAIDs) that primarily target COX enzymes [51,52,53]. Moreover, C_17_-PAs can modulate inflammatory signaling pathways, such as nuclear factor kappa B (NF-κB) and mitogen-activated protein kinase (MAPK) cascades [54]. NF-κB is a central regulator of inflammation, controlling the transcription of genes encoding pro-inflammatory cytokines and adhesion molecules [48]. Falcarinol has been shown to inhibit NF-κB activation by preventing its nuclear translocation, thus reducing the expression of inflammatory genes [55,56]. Additionally, C_17_-PAs downregulate MAPK signaling pathways, particularly the p38 and JNK pathways, further attenuating inflammatory responses [3]. These multi-targeted mechanisms make C_17_-PAs promising candidates for the development of natural anti-inflammatory therapeutics.

### 3.3. Antimicrobial and Antifungal Activities of C_17_-Polyacetylenes

C_17_-PAs also exhibit broad-spectrum antimicrobial properties, demonstrating activity against both gram-positive and gram-negative bacteria, as well as various fungal pathogens [57,58,59,60]. For instance, falcarindiol and falcarinol have been shown to inhibit the growth of bacterial species such as *Staphylococcus aureus*, *Escherichia coli* and *Pseudomonas aeruginosa* [61,62,63]. The antimicrobial effects are primarily attributed to the disruption of bacterial cell membranes, leading to increased permeability and eventual cell lysis. This membrane-targeting mechanism makes C_17_-PAs effective against antibiotic-resistant strains, highlighting their potential in the fight against multidrug-resistant pathogens.

In addition to their antibacterial properties, C_17_-PAs exhibit antifungal activity against phytopathogens and human fungal pathogens [30,64]. Falcarinol and its derivatives inhibit the growth of Candida albicans and *Aspergillus fumigatus* by interfering with ergosterol biosynthesis, which is essential for fungal membrane integrity [65,66]. This leads to membrane destabilization and fungal cell death. Furthermore, C_17_-PAs have been found to disrupt fungal biofilm formation, which is a key factor in chronic infections and antifungal resistance [67,68]. By preventing biofilm establishment, these compounds enhance the efficacy of conventional antifungal agents. These compounds also exhibit synergistic effects with existing antibiotics and antifungal drugs. For instance, combining falcarindiol with amphotericin B or fluconazole enhances the antifungal activity against resistant fungal strains [2,69]. This synergy suggests that C_17_-PAs could be used as adjuvants to existing antimicrobial therapies, potentially reducing the required dosage of conventional drugs and minimizing side effects. Additionally, the relatively low toxicity of C_17_-PAs at therapeutic doses makes them attractive candidates for future pharmaceutical development.

### 3.4. Neuroprotective and Neuromodulatory Effects of C_17_-Polyacetylenes

Beyond their anticancer, anti-inflammatory and antimicrobial properties, C_17_-polyacetylenes have shown neuroprotective and neuromodulatory potential [12,70]. At lower concentrations, these compounds enhance neuronal survival and protect against oxidative stress-induced damage [71]. Falcarinol, in particular, has been shown to activate the Nrf2 pathway in neuronal cells, increasing the expression of antioxidant enzymes such as superoxide dismutase (SOD) and catalase [52,53]. This reduces ROS accumulation, protecting neurons from oxidative stress and apoptosis. Such antioxidant effects are particularly relevant for neurodegenerative diseases, including Alzheimer’s and Parkinson’s disease, where oxidative damage plays a critical role. Furthermore, C_17_-PAs modulate neurotransmitter systems, particularly the serotonergic and dopaminergic pathways [72,73,74]. Falcarinol has been reported to enhance serotonergic signaling by increasing serotonin receptor sensitivity, which may have antidepressant-like effects [75]. This neuromodulatory activity is of particular interest in the context of mood disorders and cognitive decline [76]. Additionally, falcarinol interacts with endocannabinoid receptors, exerting anxiolytic and antinociceptive effects [77,78]. These interactions suggest potential therapeutic applications C_17_-PA in managing chronic pain and anxiety-related conditions.

However, it is important to note that C_17_-PAs can exhibit neurotoxic effects at higher concentrations, causing membrane disruption and neuronal damage [7,79]. This dose-dependent duality highlights the need for precise dosage optimization to harness their neuroprotective benefits while avoiding toxicity. Further in vivo studies and clinical trials are necessary to validate their efficacy and safety in treating neurological disorders. Overall, the dual nature of C_17_-PAs—acting as both protective phytochemicals and context-dependent toxins—underscores the need for further research into their biosynthetic pathways and in vivo efficacy to fully exploit their benefits in medicine and agriculture.

## 4. Current Understanding of the C_17_-Polyacetylene Biosynthetic Pathway in Carrot

The understanding of C_17_-PA biosynthesis is best studied in carrot, which serves as a well-established model system for unraveling the underlying metabolic and regulatory mechanisms. Despite their recognized roles in plant defense and human health, the biosynthetic route leading to these compounds remains only partially resolved. Recent advances in plant genomics, transcriptomics and metabolite profiling have begun to shed light on the complex enzymatic machinery, genetic regulation and environmental modulation involved in their production. In the following subsections, we explain the key enzymatic steps, candidate gene families and external factors influencing the biosynthesis and accumulation of C_17_-PAs in carrots.

### 4.1. Enzymatic Mechanisms

The biosynthesis of C_17_-PAs in carrots initiates with linoleic acid (C18:2), a prevalent polyunsaturated fatty acid, which undergoes specialized enzymatic transformations to yield acetylenic compounds with distinct bioactivities [18]. Central to this pathway are FAD2-like acetylenases, enzymes within the fatty acid desaturase family, which catalyze the insertion of triple bonds—a defining feature of PAs (Figure 3). These enzymes diverge functionally from canonical desaturases; for instance, Δ12-acetylenases convert linoleic acid to crepenynic acid (C18:3Δ9,12,14) through a series of hydroxylation and dehydrogenation reactions [18]. A subsequent desaturation step, catalyzed by bifunctional FAD2 enzymes, leads to the formation of dehydrocrepenynic acid (C18:4Δ9,12,14,16), which serves as the immediate precursor to C_17_-PAs (Table 1). This is followed by chain shortening via β-oxidation, reducing the C_18_ backbone to a C_17_ carbon skeleton. Further functionalization—typically through regioselective hydroxylation and acetylation—produces bioactive derivatives such as falcarinol and falcarindiol, which are implicated in plant defense and human health [80].

Recent heterologous expression studies have substantiated the catalytic function of carrot FAD2 paralogs in producing acetylenic fatty acids when introduced into non-native systems, confirming their biochemical specificity [81]. For example, studies have shown that DcFAD2-11 and DcFAD2-6 are key desaturase paralogs capable of catalyzing both acetylenation and dehydrogenation, enabling the sequential transformation from linoleic acid to crepenynic and dehydrocrepenynic acids [81]. These findings are supported by structural studies indicating that single amino acid substitutions near conserved histidine motifs significantly influence FAD2 catalytic specificity, reflecting their functional plasticity [13,20].

### 4.2. Genetic and Tissue-Specific Regulation

Recent advances in genetic mapping have provided compelling evidence for the genetic control of C_17_-PAs biosynthesis in carrots. Notably, quantitative trait loci (QTLs) associated with the accumulation of key polyacetylenes—falcarinol and falcarindiolhave been identified on chromosomes 4 and 9 of the carrot genome [82]. These loci also coincide with genomic regions previously linked to bitterness perception, indicating a functional connection between metabolite content and sensory traits. Within these QTL intervals, candidate biosynthetic genes, including desaturases and members of the FAD2 family, were found. These genes are believed to catalyze essential steps in the conversion of linoleic acid to acetylenic intermediates, a critical part of the falcarin-type PAs pathway. Their strong co-expression with PA levels across diverse genotypes further supports their involvement in regulating the biosynthetic flux toward falcarinol and falcarindiol production.

Furthermore, cultivar-specific variations in PA content—exhibiting up to 10-fold differences across genotypes—highlight strong genetic control [20,82,83]. For instance, PA accumulation is tissue-specific, with root periderm [2] containing 3–5 times higher PA concentrations than cortex or phloem, consistent with spatially restricted enzyme activity [84,85]. The key regulatory genes, such as *DCAR_032551* (a homolog of Arabidopsis photomorphogenesis regulators), co-express with PA biosynthetic enzymes, suggesting a link between light signaling and PA production [20]. Whole-genome duplications (WGDs) in carrots have expanded gene families like JmjC and B3-domain transcription factors, which may coordinate PA biosynthesis with stress responses [86]. Recent transcriptomic and coexpression analyses have identified additional genes and gene cluster such as *DCAR_017011*, *DCAR_003420* and *DCAR_002026* that show strong periderm-specific and pathogen-induced expression, supporting their likely involvement in PA biosynthesis [20]. Metabolite profiling further confirms that falcarindiol is the dominant PA in the periderm, making up ~83% of total PAs, while falcarinol is more prominent in phloem tissues [22]. These spatial patterns align with the expression domains of FAD2-like genes, providing molecular evidence for tissue-specific regulation.

### 4.3. Environmental and Microbial Elicitaions

Environmental and developmental cues also influence PA biosynthesis. Pathogen challenges, such as fungal infections caused by Botrytis cinerea (grey mold) and *Mycocentrospora acerina* (liquorice rot), trigger PA accumulation, reinforcing their role as critical components of plant defense [84]. These bitter-tasting secondary metabolites, particularly falcarinol-type PAs, are thought to act as phytoalexins or phytoanticipins, providing an antifungal shield by disrupting fungal hyphal growth and limiting pathogen spread. For example, cultivars like ‘Bolero’ exhibited the highest tolerance to fungal infection and maintained higher PA levels after 6 weeks of cold storage, suggesting that constitutive PA accumulation in cortex and periderm enhances resistance [82]. Furthermore, fungal elicitor application has been shown to upregulate key biosynthetic genes such as *DcFAD2*-11 and *DcFAD2*-6 [87], reinforcing their inducible role in PA defense. In another study, Alternaria dauci-challenged carrot lines with high falcarindiol content showed reduced disease symptoms, linking PA accumulation to functional resistance [30]. Although PA concentrations showed a negative, non-significant correlation with disease severity, the findings suggest that higher constitutive PA levels may contribute to enhanced resistance, particularly in the cortex and periderm, supporting their defensive role against post-harvest fungal pathogens.

Moreover, soil composition, particularly nitrogen availability, influences the precursor fatty acid pools essential for PA biosynthesis, while microbial communities in the rhizosphere indirectly modulate the PA pathway and its activities [88]. Rhizosphere microorganisms, such as specific bacteria and fungi, can alter PA production by influencing plant gene expression, enzyme activity or substrate availability in the biosynthetic pathway. For instance, certain rhizobacteria produce signaling molecules, like volatile organic compounds or quorum-sensing signals, that may upregulate defense-related genes, including those involved in PA synthesis. Additionally, microbial degradation of organic matter in the rhizosphere may also modify local nutrient profiles, indirectly affecting the availability of precursors like acetyl-CoA for PA production. While rhizosphere communities are the primary microbial influencers due to their proximity to root tissues where PAs are synthesized, other soil microbial populations could also contribute by shaping broader soil chemistry, although their impact is less direct [89].

### 4.4. Knowledge Gaps in Carrot Polyacetylene Biosynthetic Pathway

Despite the genetic and biochemical insights available, several critical aspects of the C_17_-PA biosynthetic pathway remain poorly understood. Notably, the enzymes responsible for the β-oxidation step that shortens C_18_ precursors to C_17_ skeletons have not been yet identified. Likewise, the regioselective terminal functionalization steps (e.g., acetylation of falcarindiol-3-acetate) lack characterized enzymes [13]. Radiolabeling studies in *Panax ginseng* suggest limited incorporation of labeled linoleic acid into final PA products. The low incorporation of labeled linoleic acid challenges this linear assumption, implying that either the metabolic flux from linoleic acid to PA is minimal or bypassed, or that linoleic acid might not be the immediate precursor in ginseng. The findings raise the possibility that other precursors, such as acetyl-CoA units, oxylipin intermediates or branched-chain fatty acids, could feed into the polyacetylene biosynthetic pathway [90]. These observations suggest that PA biosynthesis may be species-specific and mechanisms in *Panax ginseng* may differ substantially from those in Apiaceae species such as carrot, where linoleic acid-derived pathways have stronger supporting evidence.

In addition, limited information is available regarding how transcription factors and WGD events across carrot genotypes regulate this pathway, though expanded gene families such as JmjC and B3-domain TFs have been implicated [54,86]. Another key research gap is the interplay between PA and carotenoid biosynthesis, which both draw on shared acetyl-CoA pools. This metabolic competition remains largely unexplored and could have implications for yield trade-offs under variable environmental conditions [2,17]. Further, the evolutionary basis for Apiaceae-specific FAD2 diversification, including the expansion of acetylenase subtypes, is still unclear, although recent phylogenomic work has identified potential duplication and neofunctionalization events. Moreover, thermal and photochemical instability of PAs complicates their isolation and quantification, often leading to artifactual degradation during extraction [91]. Advanced analytical methods, such as UV-spectral fingerprinting, can help mitigate this but require specialized protocols to preserve native compound structures [92,93]. Metabolic engineering efforts face hurdles due to the low substrate promiscuity of key enzymes like FAD2-like acetylenases, which show strict specificity for linoleic acid derivatives [20]. Similarly, the use of heterologous expression systems often fails to replicate the high PA yields observed in native carrot tissues, likely due to missing regulatory elements or compartmentalized precursor pools [94,95]. PA accumulation is influenced by inducible defense responses to pathogens and abiotic stressors, but the signaling molecules coordinating these responses (e.g., jasmonates vs. salicylates) are undefined [59,96].

Recent advances in transgenic approaches, such as overexpression of identified FAD2 variants in Arabidopsis, tobacco and yeast system, offer promising strategies to engineer PA-enriched carrots for enhanced disease resistance and nutritional value. However, these emerging techniques including CRISPR/Cas9 also face alternative challenges in carrot systems, such as inefficient protoplast regeneration and transgene silencing [95,97]. These limitations underscore the need for multi-omics approaches to resolve pathway topology and regulation. Also, by prioritizing enzyme characterization through activity-guided protein purification, coupled with isotopic tracer studies can aid in the validation of precursor flux, as well as address mechanistic uncertainties. Concurrently, developing stable carrot transformation systems and tissue-specific promoters would enable precise genetic dissection of PA biosynthesis, bridging the gap between pathway elucidation and agricultural application.

## 5. Methodological Challenges and Strategic Advances in Polyacetylene Pathway Elucidation

### 5.1. Methodological and Analytical Approaches

A central challenge in PA research is the thermal and photochemical instability of falcarinol-type compounds, which complicates their isolation, quantification and the reproducibility of analytical outcomes [89]. Many PAs rapidly degrade or rearrange under standard laboratory lighting or elevated temperatures, leading to artifacts that obscure true metabolic profiles. To mitigate degradation during extraction, methods such as cryogenic grinding, use of amber vials, rapid cold solvent extraction and low-light workflows have been increasingly adopted [98]. For analytical quantification, UV-spectral fingerprinting, high-performance liquid chromatography-mass spectrometry (HPLC-MS) and gas chromatography-mass spectrometry (GC-MS) remain indispensable tools [24,62,99]. These techniques allow the separation and quantification of structurally similar PAs, although isomer differentiation still requires sophisticated detectors or complementary techniques such as NMR or HR-MS/MS (Figure 4). Additionally, the use of isotopic tracers, including ^13^C- and ^2^H-labeled precursors, have emerged as promising approaches to validate precursor-product relationships and enzyme activities. These studies help trace carbon flow through the biosynthetic pathway and assess metabolic fluxes, thereby clarifying contentious steps such as the desaturation-acetylenation cascade and chain-shortening mechanisms.

### 5.2. Comparative Insights from Other Apiaceae

Comparative studies with other Apiaceae species such as celery (*Apium graveolens*) and parsley (*Petroselinum crispum*) suggest both conserved and divergent elements in C_17_-PA metabolism. For instance, while falcarinol and falcarindiol are present across these species, carrot accumulates significantly higher levels and exhibits more complex tissue-specific regulation and responsiveness to environmental cues such as temperature, wounding and pathogen challenge [22]. These differences may reflect species-specific expansions or neofunctionalization of FAD2-like acetylenases or diversification in the pool of upstream fatty acid precursors available for acetylenic modification [18]. Moreover, recent comparative transcriptomic and genomic analyses have indicated that carrot harbors a relatively expanded and transcriptionally active cluster of FAD2-like genes that are absent or poorly expressed in closely related species, hinting at a lineage-specific amplification of acetylenase capacity [18]. Despite this, the functional characterization of these homologs in non-carrot species remains limited and the evolutionary pressures driving such divergence are still poorly understood. To bridge these gaps, high-quality pan-Apiaceae comparative genomics can identify conserved and species-specific gene expansions in *FAD2*, *CYP450* and *oxylipin*-related gene families. Similarly, conducting RNA-seq under standardized biotic/abiotic stress conditions across species can also aid in understanding shared and divergent transcriptional responses involved in PA regulation.

### 5.3. Human Bioavailability and Pharmacokinetics

Recent clinical studies have shed light on the bioavailability and pharmacokinetics of key PAs. Falcarinol and falcarindiol are absorbed in the human gut, albeit with moderate efficiency due to their lipophilic nature. They undergo hepatic metabolism, resulting in hydroxylated and conjugated derivatives with reduced biological activity [3]. Plasma concentrations peak 1–2 h post-ingestion of raw carrots but decline rapidly, indicating a short half-life. Nonetheless, repeated dietary intake may help sustain bioactive levels. These findings underscore the relevance of food matrix, dose and processing on PA efficacy and motivate further studies into delivery systems and structure–activity relationships.

### 5.4. Next-Generation-Aided AI/ML Strategies

Next-generation technologies are driving strategic progress in elucidating the PA biosynthetic pathway. High-throughput RNA-seq, chromosome-level genome assemblies and co-expression network analysis have helped identify candidate genes and predict biosynthetic gene clusters [59,100]. The use of CRISPR-Cas9 systems has proven effective for functional validation, given the overcoming of low protoplast regeneration and transgene silencing efficiency in carrots [101,102]. In addition, advanced HPLC-MS/GC-MS techniques, especially in conjunction with HR-MS/MS and non-targeted metabolomics, are enabling the reconstruction of oxylipin networks that link fatty acid desaturation to PA biosynthesis [103]. Click-chemistry-based alkyne tagging also offers potential for real-time visualization of PA biosynthesis, particularly when combined with microfluidic protoplast assays and spatial metabolomics tools (Figure 4). Although AI/ML approaches—including XGBoost classifiers and similarity-based pathway predictors—have shown strong potential for identifying unknown biochemical steps, their application must be better aligned with core challenges in PA research. Integrating these tools with experimental validation pipelines, such as activity-guided protein purification and enzyme-substrate docking models, would strengthen their impact [104,105,106,107].

## 6. Conclusions and Future Perspectives

C_17_-polyacetylenes, particularly falcarinol and falcarindiol in carrots, are notable for their bioactivities, including anti-inflammatory, cytotoxic and anticancer effects. These bioactive compounds are produced predominantly by plants in the Apiaceae, Araliaceae and Asteraceae families [2,15], with carrots exhibiting distinct metabolic regulation and higher accumulation compared to related species such as celery and parsley [22]. While core biosynthetic steps such as acetylenation are well-characterized, key mechanisms—especially chain shortening and terminal modifications—remain unresolved [17]. Recent studies have advanced our understanding of gene clusters and environmental regulation, while clinical findings highlight the bioavailability and pharmacokinetics of falcarinol-type PAs [7]. These compounds act as PPARγ ligands, further underscoring their pharmaceutical potential [3]. To fully exploit these benefits, integrative strategies that combine enzyme discovery, multi-omics profiling and AI-driven modeling are essential. Concurrently, improving compound stability during extraction and delivery systems will support translational applications in functional foods and nutraceuticals. A deeper understanding of stress-responsive regulation will also inform the development of resilient, biofortified carrot cultivars. Ultimately, bridging current knowledge gaps—especially regarding spatial biosynthesis, regulatory networks, and interspecies differences—will enhance our understanding of plant defense, human health and the broader metabolic landscape of polyacetylenes.

## Figures and Tables

**Figure 1 cimb-47-00471-f001:**
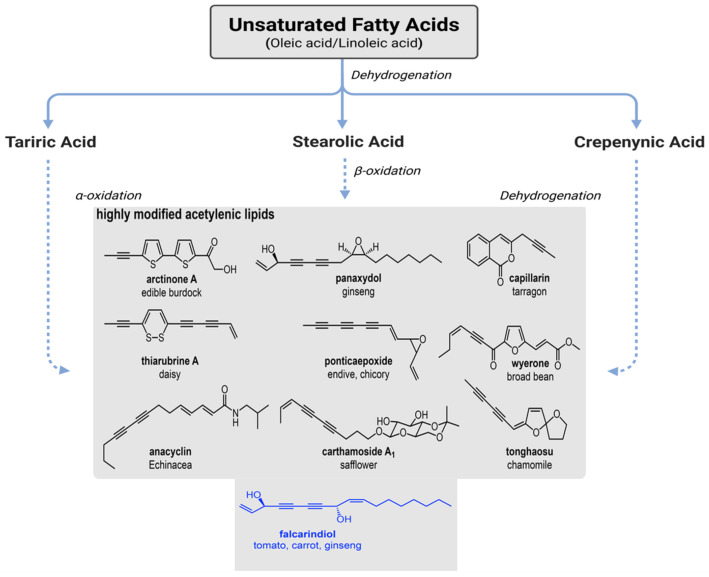
Polyacetylenic compounds derived from unsaturated fatty acids in various plant species. Their biosynthesis originates with unsaturated fatty acids (oleic acid and linoleic acid), depicted in the top box, which undergo a series of enzymatic modifications. The process begins with dehydrogenation, leading to the formation of stearolic acid, followed by β-oxidation to produce crepenynic acid and further dehydrogenation to yield highly modified acetylenic lipids. Tariric acid, derived through α-oxidation, represents an alternative entry point into the pathway. Solid arrows indicate well-established enzymatic steps, while dashed arrows denote hypothetical or less-characterized transformations based on comparative metabolic studies. The diagram illustrates the diversity of polyacetylenic compounds across plant species, with chemical structures and common names provided for each compound. Key intermediates and end products include: arcitinone A (edible burdock), thiarrubine A (daisy), panaxydol (ginseng), capillarin (tarragon), ponticaepoxide (endive, chicory), wyerone (broad bean), anacyclin (Echinacea), carthamoside A (safflower), tonghaosu (chamomile) and falcarindiol (tomato, carrot, ginseng). Each compound is associated with its primary plant source, highlighting species-specific metabolic specialization. The structure of falcarindiol, a prominent C_17_-polyacetylene, is emphasized at the bottom, reflecting its significance in carrot and related species [21].

**Figure 2 cimb-47-00471-f002:**
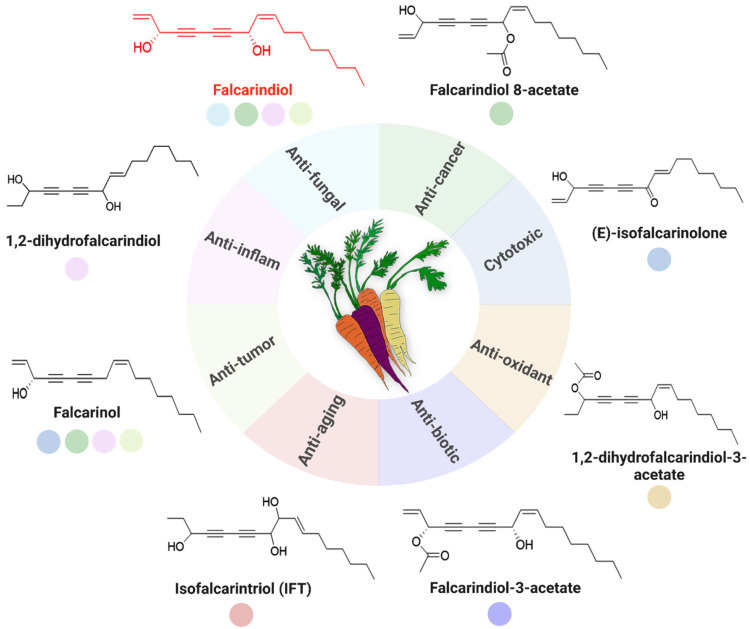
Abundance of bioactive C_17_-PAs in carrots and their multifaceted biological activities with potential health benefits. The central illustration depicts different carrot varieties and surrounding this core are multiple segments, each representing a distinct bioactivity, including anticancer (green), antifungal (light blue), cytotoxic (blue), antioxidant (yellow), antibiotic (dark blue), anti-aging (red), anti-tumor (light beige) and anti-inflammatory (pale pink). Key PA-derived compounds are depicted as colored circles adjacent to the chart, with their chemical structures shown where applicable.

**Figure 3 cimb-47-00471-f003:**
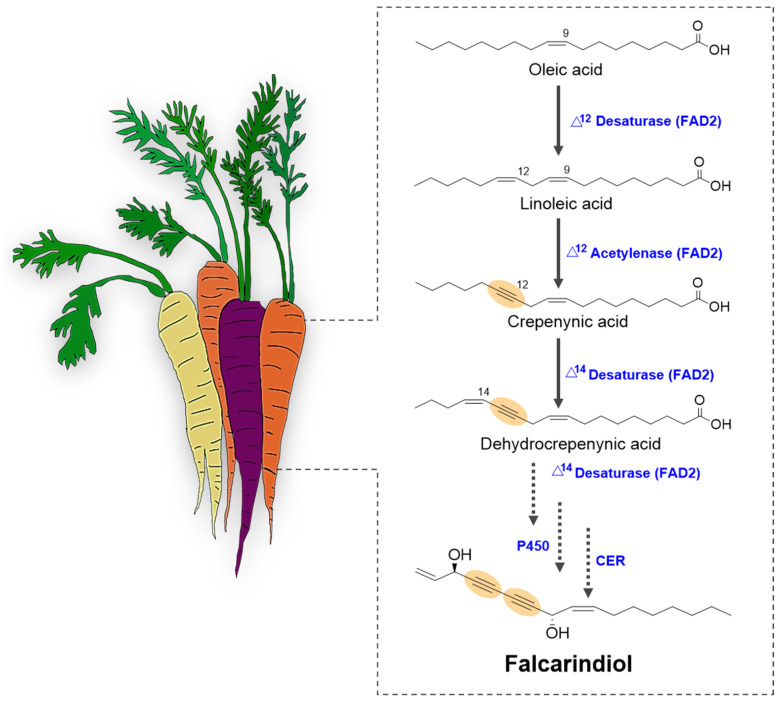
Proposed biosynthetic pathway for falcarindiol in carrots (Daucus carota). The schematic illustrates the sequential biochemical transformations leading to falcarindiol, a key C17-polyacetylenic oxylipin. Solid arrows represent enzymatic steps catalyzed by characterized enzymes, including fatty acid desaturases (FAD2), supported by experimental evidence from gene expression and biochemical assays. Dotted arrows indicate hypothesized transformations, such as hydroxylation and desaturation steps, which remain to be experimentally validated due to limited data on specific enzyme involvement or substrate specificity. Colored texts differentiate enzyme classes, intermediates are labeled with their chemical names and structures, and the function of each enzyme (double and triple bond formation) are highlighted with yellow circles in the chemical structure of the compounds. The pathway begins with linoleic acid as the primary precursor, progressing through crepenynic acid and other polyacetylenic intermediates. Hypothetical steps are based on comparative genomics and metabolomic data from Apiaceae species, with gaps highlighted for future research.

**Figure 4 cimb-47-00471-f004:**
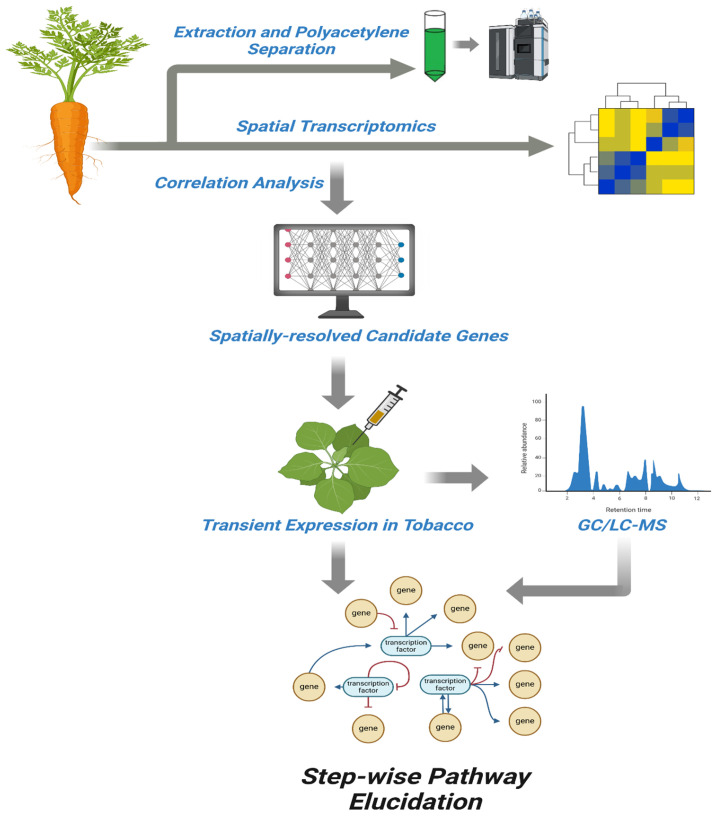
A schematic workflow depicting the integrative approach used for elucidating the biosynthetic pathway of polyacetylenes in carrot. The process begins with extraction and separation of polyacetylenes, followed by spatial transcriptomics to profile gene expression across different tissues. Correlation analysis is used to identify spatially co-expressed genes associated with polyacetylene accumulation. Candidate genes are then selected and subjected to transient expression in *Nicotiana benthamiana* to assess functional relevance. Metabolic products are analyzed via GC/LC-MS, enabling biochemical validation. The integrated data support step-wise pathway elucidation, uncovering gene regulatory networks and metabolic steps involved in polyacetylene biosynthesis.

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
