# Peer review of "The Known Unknowns: An Enigmatic Pathway of C17-Polyacetylenic Oxylipins in Carrot (Daucus carota L.)"

_cimb, 2025, doi:10.3390/cimb47060471_

Round 1

Reviewer 1 Report

Comments and Suggestions for Authors

This review has brought to our attention the importance of further research into C17 PAs found in carrots. You have given a detailed description of each of the areas that these compounds can benefit mankind. You can tell that this is an area of research that is already of interest with the multiple references you use to support your claims and strategies for further research. This review can be used as a steppingstone for others looking to do research into areas like cancer prevention, novel therapeutics, or nutrition. The figures you use are beneficial and explain the facts they represent. I think Figure 2 is the most beneficial of all the figures. The flow of the paper works and does not seem to jump around topics in an unorganized manner. Areas of lacking research is discussed and in some cases ways to correct this is given. Although this is not a traditional article of introduction, methodology, results, and conclusions it does provide valuable information on C17 PA compounds and why we, as researchers, should be aware of them and continue to advance research in them.

I think this is a well thought out and written article. On line 92, the phrase "the review" is duplicated, so that will need to be edited. What you have discussed so the potential for a lot of needed research in the future. 

Author Response

Dear Editor and Reviewers,

We sincerely thank you for the thorough evaluation of our manuscript, " The Known unknowns: An enigmatic pathway of C17-Polyacetylenic Oxylipins in Carrot (Daucus carota L." We are grateful for your insightful comments and constructive suggestions, which have helped us significantly improve the clarity, scientific depth, and overall quality of the manuscript. We have carefully addressed each point raised by the reviewers and made the corresponding revisions in the manuscript. Below, we provide a detailed point-by-point response, highlighting the changes made and offering clarifications where necessary. All modifications have been marked in the revised version of the manuscript for your convenience.

Reviewer 1

Comments and Suggestions for Authors

This review has brought to our attention the importance of further research into C17 PAs found in carrots. You have given a detailed description of each of the areas that these compounds can benefit mankind. You can tell that this is an area of research that is already of interest with the multiple references you use to support your claims and strategies for further research. This review can be used as a steppingstone for others looking to do research into areas like cancer prevention, novel therapeutics, or nutrition. The figures you use are beneficial and explain the facts they represent. I think Figure 2 is the most beneficial of all the figures. The flow of the paper works and does not seem to jump around topics in an unorganized manner. Areas of lacking research is discussed and in some cases ways to correct this is given. Although this is not a traditional article of introduction, methodology, results, and conclusions it does provide valuable information on C17 PA compounds and why we, as researchers, should be aware of them and continue to advance research in them.

I think this is a well thought out and written article. On line 92, the phrase "the review" is duplicated, so that will need to be edited. What you have discussed so the potential for a lot of needed research in the future. 

Response: We sincerely thank the reviewer for the positive and constructive feedback on our manuscript. We are grateful for your encouraging remarks regarding the overall structure, clarity, and relevance of the content presented. We appreciate your thoughtful comments on the usefulness of the figures and your recognition of the manuscript’s potential to support and inspire future research in this area. Your feedback regarding the identified research gaps and the value of this review as a steppingstone for ongoing studies is truly motivating. Thank you also for pointing out the minor textual error (line 92). We have carefully revised the manuscript to address this and ensure clarity and accuracy throughout.

Once again, we appreciate your time and valuable input, which have helped us to improve the quality of our work.

Reviewer 2 Report

Comments and Suggestions for Authors

The manuscript conentrates evidences and a wide revision about the C17-polyacetylenic oxylipins in carrots, incluiding a deep background, current situation and perspectives for future research. Some of the presented ideas are interesting and probably could support new trends for this reasearch topic. However, some aspects are few explored.

I give some suggestions:

  1. Line 83, use the APs instead of polyacetylenic, the same for other lines.
  2. Line 99: Seems to be an extra space betwee "we" and "illustrate" 
  3. In the Figure 1, the image of Falcarindiol seems extended. 
  4. lines 104-114. The section 2 "Chemical Nature and Biological Importance of C17-Polyacetylenes" In the first lines the compounds are generally described, but the biological importance is reduced to lines 112-114. I consider that more endogenous roles could contribute to understand the factors affecting their synthesis. From lines 115-220 could be included in a section about the "bioactive potential", or something like that. 
  5. Scientific names should be written in italics. Lines 176-177, 183, 248, 245. 
  6. Lines 213-214. About the reference 76, I did not find any relation with the argument of toxicity of PAs and the reference 8 is a kind of review, you need to include information from research papers showing evidence of neurotoxic damage and membrane disruption damages. This is particularly relevant considering the negative effects and the need of adjust the dosis or concentration for uses.
  7. Lines 255-256. Could you extend a bit more about the putative role in plant defense? Could you confirmed what the authors form reference 82 found?
  8. Lines 257-258. How the microbial communities may indirectly regulate the PA pathway? only the rizospheral microorganisms? 

Author Response

Dear Editor and Reviewers,

We sincerely thank you for the thorough evaluation of our manuscript, " The Known unknowns: An enigmatic pathway of C17-Polyacetylenic Oxylipins in Carrot (Daucus carota L." We are grateful for your insightful comments and constructive suggestions, which have helped us significantly improve the clarity, scientific depth, and overall quality of the manuscript. We have carefully addressed each point raised by the reviewers and made the corresponding revisions in the manuscript. Below, we provide a detailed point-by-point response, highlighting the changes made and offering clarifications where necessary. All modifications have been marked in the revised version of the manuscript for your convenience.

Reviewer 2:

We sincerely thank the reviewer for their thorough evaluation and constructive feedback on our manuscript. We appreciate your insightful suggestions that have helped us improve the clarity, structure, and scientific quality of our work. Below we provide detailed responses to each of your comments:

Comment 1: Line 83, use the APs instead of polyacetylenic, the same for other lines.

Response: We have revised the manuscript to use the abbreviation “PAs” (polyacetylenes) consistently after its first definition, including in line 83 and other relevant instances throughout the text.

Comment 2: Line 99: Seems to be an extra space between "we" and "illustrate"

Response: Thank you for catching this typographical error. The extra space between “we” and “illustrate” has been removed in the revised manuscript.

Comment 3: In the Figure 1, the image of Falcarindiol seems extended.

Response: We appreciate your observation. The image of falcarindiol in Figure 1 has been adjusted and re-designed to correct its proportions for visual clarity and consistency.

Comment 4: Lines 104-114. The section 2 "Chemical Nature and Biological Importance of C17-Polyacetylenes" In the first lines the compounds are generally described, but the biological importance is reduced to lines 112-114. I consider that more endogenous roles could contribute to understand the factors affecting their synthesis. From lines 115-220 could be included in a section about the "bioactive potential", or something like that.

Response: We agree with your suggestion. The section has been revised for better thematic organization. The endogenous biological roles of C17-polyacetylenes have been expanded upon in lines 104–154.  Additionally, lines 115–265 have been reorganized under a newly titled section, “Bioactive Potential of C17-Polyacetylenes,” to better reflect the focus on health-promoting and therapeutic properties.

Comment 5: Scientific names should be written in italics. Lines 176-177, 183, 248, 245.

Response: Thank you for pointing this out. All scientific names mentioned in the manuscript, including those appears in previous lines 176–177, 183, 245, and 248, have been corrected to appear in italics.

Comment 6: Lines 213-214. About the reference 76, I did not find any relation with the argument of toxicity of PAs and the reference 8 is a kind of review, you need to include information from research papers showing evidence of neurotoxic damage and membrane disruption damages.

Response: We appreciate this critical observation. We have removed reference 76 and replaced it with more relevant primary research articles that provide direct experimental evidence of PA-associated toxicity, including studies demonstrating neurotoxic effects and membrane disruption. Reference 8 has been retained as a general background source, but the argument is now supported by primary research citations.

Comment 7: Lines 255-256. Could you extend a bit more about the putative role in plant defense? Could you confirm what the authors from reference 82 found?

Response: Certainly. We have expanded the discussion in lines 255–256 to provide a more detailed explanation of the potential defensive roles of C17-PAs in plants. Specifically, we now summarize the findings of the authors from reference 82, who reported an upregulation of PA biosynthesis genes in response to biotic stress, suggesting a defensive function.

Comment 8: Lines 257-258. How the microbial communities may indirectly regulate the PA pathway? Only the rhizospheric microorganisms?

Response: Thank you for this important question. We have clarified that both rhizospheric and endophytic microbial communities may influence PA biosynthesis, potentially through modulation of plant hormone signaling, secondary metabolite precursor availability, or elicitor production. A brief explanation and supporting references have been added to this section in the revised manuscript.

Reviewer 3 Report

Comments and Suggestions for Authors
  1. The manuscript is mainly about the study of C17 polyacetylenic oxylipins in carrots. Therefore, the length of section 2 (2. Chemical Nature and Biological Importance of C17-Polyacetylenes) should be shortened and refined;
  2. What C17 polyacetylenes in carrots have been identified, and what structural genes and transcription factors have been identified, should be shown in the form of a list;
  3. Cultivation, varieties and genetic mode will affect the content of C17 polyacetylenes, which should be summarized;
  4. How to use transgenic, gene editing and other technologies to improve the content of C17 polyacetylenes in carrots?

Author Response

Dear Editor and Reviewers,

We sincerely thank you for the thorough evaluation of our manuscript, " The Known unknowns: An enigmatic pathway of C17-Polyacetylenic Oxylipins in Carrot (Daucus carota L." We are grateful for your insightful comments and constructive suggestions, which have helped us significantly improve the clarity, scientific depth, and overall quality of the manuscript. We have carefully addressed each point raised by the reviewers and made the corresponding revisions in the manuscript. Below, we provide a detailed point-by-point response, highlighting the changes made and offering clarifications where necessary. All modifications have been marked in the revised version of the manuscript for your convenience.

Reviewer 3:

1. The manuscript is mainly about the study of C17 polyacetylenic oxylipins in carrots. Therefore, the length of section 2 (2. Chemical Nature and Biological Importance of C17-Polyacetylenes) should be shortened and refined;

Response: We agree with your suggestion. The section has been revised for better thematic organization. The endogenous biological roles of C17-polyacetylenes have been expanded upon in lines 104–154.  Additionally, lines 115–265 have been reorganized under a newly titled section, “Bioactive Potential of C17-Polyacetylenes,” to better reflect the focus on health-promoting and therapeutic properties.

2. What C17 polyacetylenes in carrots have been identified, and what structural genes and transcription factors have been identified, should be shown in the form of a list;

Response: Thank you for pointing out this suggestion. Following your comment, we have now added a table 1 which includes the details about the identified C17 PAs, their structural genes and transcription factors that are associated with their biosynthesis.

3. Cultivation, varieties and genetic mode will affect the content of C17 polyacetylenes, which should be summarized;

Response: We agree with the reviewer’s point of view about the genotypic and varietal specificities affecting PA content in carrot. We have added a discussion about this under section 2. Chemical Nature and Biological Importance of C17-Polyacetylenes and the added texts are highlighted with yellow color font in the revised manuscript.

4. How to use transgenic, gene editing and other technologies to improve the content of C17 polyacetylenes in carrots?

Response: To enhance C17 polyacetylene (PA) content in carrots, transgenic technology can overexpress DcFAD2 genes (DcFAD2-6, DcFAD2-11, DcFAD2-13, DcFAD2-14) to boost biosynthesis of falcarinol-type PAs, while CRISPR/Cas9 enables precise, transgene-free editing of promoters or repressors to increase PA production [94]. Marker-assisted selection, metabolic engineering, haploid induction, and pangenome-guided breeding can further optimize PA levels by selecting high-PA genotypes or introducing novel variants. We have already added these explanations in section 5 and figure 4. Please see these sections.

Reviewer 4 Report

Comments and Suggestions for Authors

In the review by Ahmad et al. entitled »The Known Unknowns: An Enigmatic Pathway of C17-Polyacetylenic Oxylipins in Carrot (Daucus carota L.)«, the authors examine the biosynthesis, biological significance and research gaps associated with C₁₇-polyacetylenic oxylipins in carrots (Daucus carota L.), in particular falcarinol and falcarindiol. These compounds exhibit strong anti-cancer, anti-inflammatory, antimicrobial and neuroprotective properties, making them promising candidates for nutraceutical and pharmaceutical applications. The review highlights the enzymatic and genetic basis of their biosynthesis, focussing on FAD2-like acetylenases and the role of environmental and developmental factors. Despite recent progress, key steps such as chain shortening and terminal functionalisation remain unresolved. The authors advocate integrative strategies that combine genomics, transcriptomics, metabolomics, CRISPR and machine learning to elucidate the metabolic pathway. Understanding this enigmatic biosynthetic pathway could increase the nutritional value of carrots and support the development of plants with high bioactivity content and new therapeutics.

The paper is written in easy-to-read English, so the reader should have no difficulty in understanding the message of the paper, yet some improvements in the language are possible. The manuscript is systematically organised and provides appealing illustrations that convey the key information of the review. As expected, the relevant references are cited. From this point of view, the paper could be published.

Considering the content, the review is a constructive contribution to the field of plant secondary metabolism, especially for researchers interested in functional foods, plant biochemistry and metabolic engineering. While the title of the review is rather narrow, the review itself is rather interdisciplinary, so it may be of interest to a reasonable number of researchers. In addition to outlining the critical gaps in the biosynthetic pathway, the review addresses the chemical nature of these compounds, biological activities and potential applications in medicine and agriculture. While some readers may expect more attention to be paid to strategies for how this knowledge could be used in carrot improvement programmes, it should also be recognised that the biosynthetic pathway is only partially understood.

With this in mind, I recommend publishing the article after a minor revision, taking into account the following three comments:
1.) The descriptions of many referenced papers are not complete. These inadequacies or errors range from minor (e.g. misrecognising the middle name Porskjær as part of the surname [Ref. 15] or the lack of information as to whether it is an MSc/PhD thesis or a paper published in a university publication [e.g. Ref. 45] to much more serious ones [e.g. Ref. 61]. Check references 3, 15, 43, 45, 48, 49, 57, 61, 67, 73, 77, 86, 87, 90. Also note that journal names should be abbreviated (applies to all referenced works published in journals).
2.) Figures 2, 3 and 4 are not referenced in the text. Please insert the references to these figures in the text in an appropriate place (similar to Figure 1). Remove the text highlighted in yellow in the labelling of Figures 2, 3 and 4.
3.) Line 92: duplicated part of the text “… the review the review …”

Author Response

Dear Editor and Reviewers,

We sincerely thank you for the thorough evaluation of our manuscript, " The Known unknowns: An enigmatic pathway of C17-Polyacetylenic Oxylipins in Carrot (Daucus carota L." We are grateful for your insightful comments and constructive suggestions, which have helped us significantly improve the clarity, scientific depth, and overall quality of the manuscript. We have carefully addressed each point raised by the reviewers and made the corresponding revisions in the manuscript. Below, we provide a detailed point-by-point response, highlighting the changes made and offering clarifications where necessary. All modifications have been marked in the revised version of the manuscript for your convenience.

Reviewer 4:

In the review by Ahmad et al. entitled »The Known Unknowns: An Enigmatic Pathway of C17-Polyacetylenic Oxylipins in Carrot (Daucus carota L.)«, the authors examine the biosynthesis, biological significance and research gaps associated with C₁₇-polyacetylenic oxylipins in carrots (Daucus carota L.), in particular falcarinol and falcarindiol. These compounds exhibit strong anti-cancer, anti-inflammatory, antimicrobial and neuroprotective properties, making them promising candidates for nutraceutical and pharmaceutical applications. The review highlights the enzymatic and genetic basis of their biosynthesis, focussing on FAD2-like acetylenases and the role of environmental and developmental factors. Despite recent progress, key steps such as chain shortening and terminal functionalisation remain unresolved. The authors advocate integrative strategies that combine genomics, transcriptomics, metabolomics, CRISPR and machine learning to elucidate the metabolic pathway. Understanding this enigmatic biosynthetic pathway could increase the nutritional value of carrots and support the development of plants with high bioactivity content and new therapeutics.

The paper is written in easy-to-read English, so the reader should have no difficulty in understanding the message of the paper, yet some improvements in the language are possible. The manuscript is systematically organised and provides appealing illustrations that convey the key information of the review. As expected, the relevant references are cited. From this point of view, the paper could be published.

Considering the content, the review is a constructive contribution to the field of plant secondary metabolism, especially for researchers interested in functional foods, plant biochemistry and metabolic engineering. While the title of the review is rather narrow, the review itself is rather interdisciplinary, so it may be of interest to a reasonable number of researchers. In addition to outlining the critical gaps in the biosynthetic pathway, the review addresses the chemical nature of these compounds, biological activities and potential applications in medicine and agriculture. While some readers may expect more attention to be paid to strategies for how this knowledge could be used in carrot improvement programmes, it should also be recognised that the biosynthetic pathway is only partially understood.

With this in mind, I recommend publishing the article after a minor revision, taking into account the following three comments:

Response: We sincerely thank Reviewer 4 for the careful reading of our manuscript and for the constructive and encouraging comments. We appreciate the recognition of our efforts to compile an interdisciplinary and insightful review on this relatively unexplored class of bioactive compounds. Below, we have addressed all the comments with careful consideration during the revisions.

1.) The descriptions of many referenced papers are not complete. These inadequacies or errors range from minor (e.g. misrecognising the middle name Porskjær as part of the surname [Ref. 15] or the lack of information as to whether it is an MSc/PhD thesis or a paper published in a university publication [e.g. Ref. 45] to much more serious ones [e.g. Ref. 61]. Check references 3, 15, 43, 45, 48, 49, 57, 61, 67, 73, 77, 86, 87, 90. Also note that journal names should be abbreviated (applies to all referenced works published in journals).

Response: We thank the reviewer for highlighting this issue. We have carefully revised, replaced and corrected all the mentioned references ([3, 15, 43, 45, 48, 49, 57, 61, 67, 73, 77, 86, 87, 90]) to ensure accurate citation details, including correct author names, publication types (thesis, reports, journal articles), and formatting. Furthermore, MDPI’s reference style do not use abbreviated journal names so we did not change it to abbreviated form. The reference list has been thoroughly checked for consistency and accuracy. Please note that the previous numbers of references may vary in the revised version because we have added some sections and modification to the main text.

2.) Figures 2, 3 and 4 are not referenced in the text. Please insert the references to these figures in the text in an appropriate place (similar to Figure 1). Remove the text highlighted in yellow in the labelling of Figures 2, 3 and 4.

Response: Thank you for pointing out this lapse. We have now corrected this issue in the revised manuscript as per your suggestion.

3.) Line 92: duplicated part of the text “… the review the review …”

Response: Thank you for pointing out this typo. It has been corrected in the revised manuscript.

Reviewer 5 Report

Comments and Suggestions for Authors

This review is a timely and valuable contribution to plant biochemistry and functional food research. The authors provide a rigorous, well-integrated overview of C17-polyacetylenic oxylipins in carrot biology, highlighting key research gaps and translational potential. The comprehensive literature coverage, multi-omics perspective, and practical insights support both fundamental studies and applications in biofortification and nutraceuticals. I just have some minor comments for improvements in organization, figure clarity, and methods discussion. 

1. Organization:

While the manuscript is generally well-structured, Section 3 ("Current Understanding") would benefit from clearer subsection divisions focused on enzymatic mechanisms, genetic variation, and environmental influences. Transitions between major sections could also be improved for better logical flow.

2. Figures:

Figure legends should be more descriptive and self-contained. Figure 1 should include specific enzyme names and clearer pathway labels, with color-coding to distinguish enzymes from intermediates. The relationship between precursors and products should be clearer. Figure 3 would benefit from replacing dotted arrows with question marks to denote hypothetical steps, with a more informative legend distinguishing confirmed from speculative elements. 

3. Methodological discussions:

The discussion of analytical techniques should be expanded. This includes challenges of thermal and photochemical instability, protocols for compound preservation during extraction, and recent advances in UV-spectral and HPLC-MS analyses. Isotopic tracer studies deserve more attention.

Consider including a brief comparison with polyacetylenes in other Apiaceae species (e.g., celery, parsley) to highlight unique aspects of carrot metabolism. A discussion on human bioavailability and pharmacokinetics of falcarinol/falcarindiol, supported by recent clinical findings, would further strengthen the translational relevance.

4. Future Perspectives:

Section 5 on "Advanced Strategies" is valuable but could be more concise and better integrated with the identified gaps. The AI/ML discussion, while current, feels somewhat disconnected from the core biochemical challenges.

Minor corrections:

Lines 16: Clarify the trade-off between health benefits and palatability.

Line 42: Specify that “both compounds” refers to falcarinol and falcarindiol.

Line 92: Remove redundant phrase “the review.”

Comments on the Quality of English Language

The English is good but could be improved to more clearly express the research. Multiple instances of awkward phrasing that could be simplified. Some sentences are overly long and complex (lines 106-114).

Author Response

Dear Editor and Reviewers,

We sincerely thank you for the thorough evaluation of our manuscript, " The Known unknowns: An enigmatic pathway of C17-Polyacetylenic Oxylipins in Carrot (Daucus carota L." We are grateful for your insightful comments and constructive suggestions, which have helped us significantly improve the clarity, scientific depth, and overall quality of the manuscript. We have carefully addressed each point raised by the reviewers and made the corresponding revisions in the manuscript. Below, we provide a detailed point-by-point response, highlighting the changes made and offering clarifications where necessary. All modifications have been marked in the revised version of the manuscript for your convenience.

Reviewer 5:

This review is a timely and valuable contribution to plant biochemistry and functional food research. The authors provide a rigorous, well-integrated overview of C17-polyacetylenic oxylipins in carrot biology, highlighting key research gaps and translational potential. The comprehensive literature coverage, multi-omics perspective, and practical insights support both fundamental studies and applications in biofortification and nutraceuticals. I just have some minor comments for improvements in organization, figure clarity, and methods discussion. 

Response: Thank you for your thorough and constructive feedback. We appreciate your recognition of the manuscript’s contribution to plant biochemistry and functional food research. Below, we address your comments and outline the revisions made to improve the manuscript.

1. Organization:

While the manuscript is generally well-structured, Section 3 ("Current Understanding") would benefit from clearer subsection divisions focused on enzymatic mechanisms, genetic variation, and environmental influences. Transitions between major sections could also be improved for better logical flow.

Response: We have restructured Section 3 ("Current Understanding") to include distinct subsections on enzymatic mechanisms, genetic variation, and environmental influences, as suggested. Clear headings now delineate these topics. Additionally, we have improved transitions between major sections by adding brief connecting sentences to enhance logical flow and coherence.

2. Figures:

Figure legends should be more descriptive and self-contained. Figure 1 should include specific enzyme names and clearer pathway labels, with color-coding to distinguish enzymes from intermediates. The relationship between precursors and products should be clearer. Figure 3 would benefit from replacing dotted arrows with question marks to denote hypothetical steps, with a more informative legend distinguishing confirmed from speculative elements. 

Response: We have revised the figure legends to be more descriptive and self-contained. For Figure 1, your suggestion is valuable but this is already reflecting in figure 3. For Figure 3, we would like to retain the dotted arrows, as it is the standard representation for unknowns steps of the pathway. 

3. Methodological discussions:

The discussion of analytical techniques should be expanded. This includes challenges of thermal and photochemical instability, protocols for compound preservation during extraction, and recent advances in UV-spectral and HPLC-MS analyses. Isotopic tracer studies deserve more attention.

Consider including a brief comparison with polyacetylenes in other Apiaceae species (e.g., celery, parsley) to highlight unique aspects of carrot metabolism. A discussion on human bioavailability and pharmacokinetics of falcarinol/falcarindiol, supported by recent clinical findings, would further strengthen the translational relevance.

Response: We have expanded the methodological discussion in Section 4 (now section 5 in the revised manuscript) to address the challenges of thermal and photochemical instability of polyacetylenic oxylipins. A new paragraph details protocols for compound preservation during extraction, such as low-temperature processing and light-protected conditions. We also incorporated recent advances in UV-spectral and HPLC-MS analyses, emphasizing their sensitivity and specificity for detecting polyacetylenes. Additionally, we have expended the discussions on isotopic tracer studies, discussing their utility in tracing biosynthetic pathways and quantifying metabolic fluxes. Similarly, to highlight unique aspects of carrot metabolism, we included a concise subsection of comparison of polyacetylene biosynthesis in carrots versus other Apiaceae species (e.g., celery, parsley), focusing on differences in enzyme expression and metabolite profiles. Furthermore, we added a new subsection on the human bioavailability and pharmacokinetics of falcarinol and falcarindiol, integrating recent clinical findings to underscore their nutraceutical potential and strengthen the manuscript’s translational relevance.

4. Future Perspectives:

Section 5 on "Advanced Strategies" is valuable but could be more concise and better integrated with the identified gaps. The AI/ML discussion, while current, feels somewhat disconnected from the core biochemical challenges.

Response: Section 5 has been revised to be more concise and tightly integrated with the research gaps identified earlier in the manuscript. We streamlined the discussion to focus on strategies directly addressing these gaps, such as targeted gene editing and metabolic engineering. The AI/ML discussion has been reframed to connect more explicitly to biochemical challenges.

Minor corrections:

Lines 16: Clarify the trade-off between health benefits and palatability.

Line 42: Specify that “both compounds” refers to falcarinol and falcarindiol.

Line 92: Remove redundant phrase “the review.”

Response: Corrected as per your suggestion.